# PRC1-mediated epigenetic programming is required to generate the ovarian reserve

Mengwen Hu [1,2], Yu-Han Yeh [1,2], Yasuhisa Munakata [1,2], Hironori Abe [1,2], Akihiko Sakashita[2,3], So Maezawa[2,4], Miguel Vidal [5], Haruhiko Koseki [6], Neil Hunter [1,7], Richard M. Schultz[8,9] & Satoshi H. Namekawa [1,2] ✉

The ovarian reserve defines the female reproductive lifespan, which in humans spans decades due to robust maintenance of meiotic arrest in oocytes residing in primordial follicles. Epigenetic reprogramming, including DNA demethylation, accompanies meiotic entry, but the chromatin changes that underpin the generation and preservation of ovarian reserves are poorly defined. We report that the Polycomb Repressive Complex 1 (PRC1) establishes repressive chromatin states in perinatal mouse oocytes that directly suppress the gene expression program of meiotic prophase-I and thereby enable the transition to dictyate arrest. PRC1 dysfuction causes depletion of the ovarian reserve and leads to premature ovarian failure. Our study demonstrates a fundamental role for PRC1-mediated gene silencing in female reproductive lifespan, and reveals a critical window of epigenetic programming required to establish ovarian reserve.

[1] Department of Microbiology and Molecular Genetics, University of California, Davis, Davis, CA, USA. [2] Division of Reproductive Sciences, Division of Developmental Biology, Perinatal Institute, Cincinnati Children's Hospital Medical Center, Cincinnati, OH, USA. [3] Department of Molecular Biology, Keio University School of Medicine, Tokyo, Japan. [4] Department of Applied Biological Science, Faculty of Science and Technology, Tokyo University of Science, Noda, Chiba, Japan. [5] Centro de Investigaciones Biológicas Margarita Salas, Department of Cellular and Molecular Biology, Madrid, Spain. [6] Developmental Genetics Laboratory, RIKEN Center for Allergy and Immunology, Yokohama, Kanagawa, Japan. [7] Howard Hughes Medical Institute, University of California, Davis, Davis, CA, USA. [8] Department of Biology, University of Pennsylvania, Philadelphia, PA, USA. [9] Department of Anatomy, Physiology and Cell Biology, School of Veterinary Medicine, University of California, Davis, Davis, CA, USA. ✉email: snamekawa@ucdavis.edu

In female embryos, primordial germ cells (PGCs) erase prior epigenetic states to enable entry into meiosis and eventual acquisition of totipotency in the zygote. Following chromosome pairing and recombination during meiotic prophase I (MPI), oocytes arrest in a stage called dictyate, and establish a finite ovarian reserve of primordial follicles, which maintains female reproductive lifespan. In human females, oocytes can remain arrested for decades prior to growth and maturation into eggs that are capable of being fertilized and supporting early embryogenesis[1–3]. How ovarian reserves are established and maintained remains poorly understood, in particular the roles of epigenetic mechanisms, which can generate long-lasting chromatin states that define gene expression programs unique to a specific cell type[4].

Changes in chromatin states are intimately linked to changes in gene expression[5]. We suspected a role for Polycomb Repressive Complex 1, a dominant executer of Polycomb-mediated gene silencing, in establishing the ovarian reserve because of its roles in suppressing non-lineage specific genes and defining cellular identity during development[6,7]. Mammalian Polycomb proteins comprise two functionally-related major complexes—PRC1 and PRC2—that catalyze formation of two major repressive marks, monoubiquitination of H2A at lysine 119 (H2AK119ub) and trimethylation of H3 at lysine 27 (H3K27me3), respectively[8]. The catalytic core of PRC1 is formed by the E3 ubiquitin ligase RNF2 (also known as RING1B) or RING1 (also known as RING1A)[9], either of which mediates H2AK119ub, the deposition of which is essential to maintain Polycomb gene repression[10,11].

In PGCs, PRC1 directly regulates expression of germline genes to coordinate the timing of sexual differentiation[12]. In female PGCs, MPI is initiated through activation of germline genes by DNA demethylation[13–15] and release from Polycomb-mediated silencing[16–19]. In the female germline, various histone modifications and DNA methylation are established during oocyte growth[20,21]. At a later stage, PRC1 is required for the oocyte-to-embryo transcriptome transition[22,23]. Unkown is how the oocyte epigenome is regulated while the ovarian reserve is established and maintained. In this study, we sought to address how a chromatin-based cellular memory is established at the time of ovarian reserve formation because DNA methylation is not established until oocyte growth[20,21].

Here we report molecular and genetic evidence that the PRC1 establishes repressive chromatin states in perinatal oocytes, defining a dedicated epigenetic program that directs the subsequent transition from MPI to dictyate arrest and formation of the ovarian reserve.

## Results

**Perinatal Oocyte Transition (POT) in ovarian reserve formation.** In mice, several critical stages of oogenesis occur around the time of birth, including cyst breakdown, dictyate arrest, and the assembly of primordial follicles[24–27] (Fig. 1a). These morphological changes are accompanied by dramatic changes in the transcriptome[28,29]. Five distinct gene-expression clusters were revealed by clustering analysis of all expressed genes in wild-type oocytes using publicly available data[28] (Fig. 1a, Supplementary Data 1). Most importantly, this analysis detected a major transcriptome transition between postnatal days 1 and 3 (P1 to P3), as oocytes progress from MPI to dictyate arrest, marked by increased expression of 6169 genes (Clusters 1 and 2) and decreased expression of 12,317 genes (Clusters 3 and 4). We term this major transition in perinatal oocyte gene expression the Perinatal Oocyte Transition (POT). Our cluster analysis also confirmed a major transcriptome transition as oocytes initiate

growth during the primordial-to-primary follicle transition (PPT between small and large oocytes at P4 / P6)[30,31].

**PRC1 is required to generate the ovarian reserve.** Both *Ring1* and *Rnf2* belong to Cluster 4 genes, which are highly expressed prior to POT (Fig. 1a), raising the possibility that PRC1 functions in POT. To determine the function of PRC1 in POT, PRC1 function was eliminated after embryonic day 15 (E15) using *Ddx4*-Cre[32] to mutate the E3 ubiquitin ligase RNF2 on a background lacking its partially redundant paralog RING1 (RING1A) to create PRC1 conditional knockout (PRC1cKO) mice[12,33–35] (Fig. 1b).

Nuclear localization of H2AK119ub, a readout of PRC1 activity[10,36], was observed in both non-growing oocytes (Fig. 1c), from P1 pups, and growing oocytes (Supplementary Fig. 1a) from 1-month-old females. In contrast, H2AK119ub was essentially undetectable in P1 oocytes from PRC1cKO mice (Fig. 1c). At P1, when cyst breakdown and follicle assembly initiate, estimated oocyte numbers in ovaries from PRC1cKO pups were indistinguishable from those of control littermates (Fig. 1d-f). However, by P5, when the primordial follicle pool has been established, the ovaries of PRC1cKO females contained $52.76 \pm 4.75\%$ fewer oocytes than controls (PRC1ctrl; $p < 0.05$; Fig. 1d-f), indicating that the initial ovarian reserve is smaller in PRC1cKO females.

By the onset of sexual maturity (1 month), ovaries from PRC1cKO mice were degenerating, contained only a few growing follicles (Fig. 2a), and were largely devoid of primordial follicles (Fig. 2b). Ovaries from 2-month-old PRC1cKO mice contained very few healthy-looking follicles, while corpora lutea containing degenerated ooplasm (Fig. 2c, yellow arrowheads), indicative of follicle atresia, were frequently observed. Consistent with these observations, primordial follicles and all types of growing follicles were 8- and 12-fold lower in mutant ovaries compared to controls ($p < 0.01$; Fig. 2d). By 4 months of age, ovaries from PRC1cKO mice were highly degenerated without any visible follicle structures (Fig. 2e–g).

Because *Ddx4*-Cre-mediated recombination begins at E15, when oocytes are in MPI, we assessed whether the *Ddx4*-Cre mediated loss of PRC1 affects MPI progression. At P1 and P5, similar populations of germ cells reached the dictyate stage both in PRC1cKO and PRC1 control (PRC1ctrl) ovaries (Supplementary Fig. 1b, c), suggesting that MPI progression was not affected in PRC1cKO females. Further, in PRC1cKO oocytes at P1, we observed normal morphology of meiotic chromosome axes detected with SYCP3 staining and a normal basal level of DNA damage signaling detected with γH2AX staining (Supplementary Fig. 1d). The normal progression of MPI in fetal oocytes from PRC1cKO females argues against the possibility that defects in chromosome pairing and recombination are responsible for oocyte elimination during the perinatal and early postnatal period. Thus, abnormal epigenetic gene regulation is the likely cause of oocyte loss in PRC1cKO females.

**PRC1 suppresses the meiotic prophase program at the POT.** To delineate the function of PRC1 in gene expression during the POT, we performed RNA sequencing (RNA-seq) in non-growing oocytes isolated from P1 and P5 ovaries, respectively (Supplementary Fig. 2a, b). In PRC1cKO oocytes from P1 ovaries, 341 genes showed increased expression, and 150 genes had decreased expression relative to oocytes from littermate controls (Fig. 3a, Supplementary Data 2). Notably, in PRC1cKO oocytes at P5, when oocytes are fully arrested and enclosed within primordial follicles, more genes showed increased expression (3100) than decreased expression (1808) (Fig. 3b, Supplementary Data 3; gene

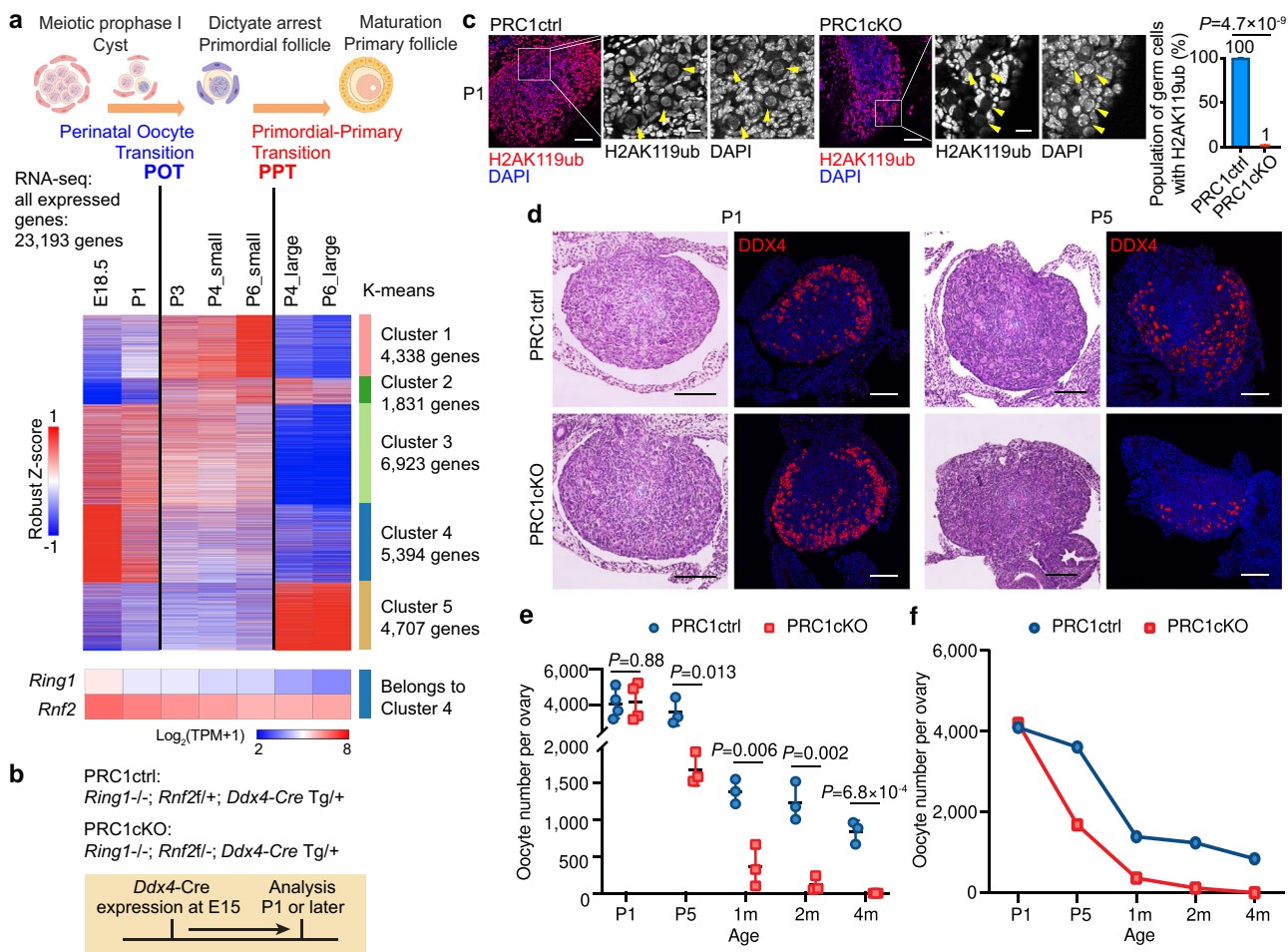

**Fig. 1 PRC1 is required for ovarian reserve establishment. a** Schematic of mouse perinatal oogenesis and dynamics of gene expression during Perinatal Oocyte Transition (POT). Heatmap showing gene expression during perinatal oogenesis detected by RNA-seq (Robust Z-score) and categorized into five k-means clusters. All 23,193 genes expressed in at least one stage (TPM > 1) were included. In wild-type, embryonic day (E) 18.5 represents oocytes in MPI; postnatal day (P) 1 and P3 represent oocytes transitioning to dictyate arrest; P4 and P6 small oocytes represent oocytes residing in primordial follicles; and P4 and P6 large oocytes represent oocytes in primary follicles after initiation of oocyte growth. PPT, primordial-to-primary follicle transition. Expression dynamics of *Ring1* and *Rnf2*, which belong to Cluster 4 during perinatal oogenesis, are shown by heatmap for $\log_2(\text{TPM}+1)$ values separately. The illustration was created with BioRender.com. **b** Schematic of mouse models and experiments. **c** Immunostaining of H2AK119ub in ovaries of PRC1cKO and a control littermate at P1. H2AK119ub is present only in somatic cells in PRC1cKO. Bars: 50 μm (10 μm in the boxed area). PRC1cKO exhibited a nearly complete loss (99%) of H2AK119ub at P1, confirming the highly efficient deletion of *Rnf2* by *Ddx4*-Cre. Data are presented as mean values ± SD. Two-tailed unpaired Student's t-tests. Three independent biological replicates were analyzed for each genotype. **d** Ovarian sections of PRC1ctrl and PRC1cKO mice at P1 and P5, respectively. The sections were stained with hematoxylin & eosin or immunostained for DDX4 (red). Bars: 100 μm. At least three mice were analyzed for each genotype at each time point, and representative images are shown. **e** Estimated numbers of oocytes per ovary from PRC1ctrl and PRC1cKO mice at P1, P5, 1 month, 2 months, and 4 months. At least three mice were analyzed for each genotype at each time point. Black bars represent mean values. Two-tailed unpaired Student's t-test. **f** Dynamics of oocyte number between P1 and 4 months of age in PRC1ctrl and PRC1cKO females. Source data are provided as a Source Data file.

expression differences at P5 are unlikely due to a developmental delay in the mutant because stage progression was not affected: Supplementary Fig. 1). Gene ontology (GO) analyses (Supplementary Fig. 2c, d) showed that genes showing increased expression in P1 and P5 PRC1cKO oocytes are enriched for processes associated with development, e.g., kidney morphogenesis, consistent with PRC1 suppressing ectopic expression of developmental programs that are not required until embryogenesis. Strikingly, genes involved in meiosis, in particular MPI, were over-represented among the genes that showed increased expression in P5 PRC1cKO oocytes (Supplementary Fig. 2d, e). GO term enrichment for genes that exhibited decreased expression in PRC1cKO oocytes at P5 were enriched in mRNA metabolic process, cell division, and chromatin organization (Supplementary Fig. 2d), suggesting a requirement for these

processes in perinatal oogenesis (enrichment was negligible in P1 oocytes, Supplementary Fig. 2c). These data imply that PRC1 is required immediately after MPI for oocytes to transcriptionally transition into arrest; in addition to a more general role in suppressing ectopic expression of inappropriate developmental programs.

To further explore the role of PRC1 in regulating transcriptional changes during POT, we first analyzed how genes in the five clusters detected in wild-type POT are regulated by PRC1. As expected, genes in Clusters 3 and 4, which show a decreased expression after POT (Fig. 1a), are up-regulated in PRC1cKO oocytes from P1 to P5 (Supplementary Fig. 2f, g). We further defined groups of differentially expressed genes (DEGs) in wild-type oocytes as POT genes and tested whether they are regulated by PRC1. Between E18.5 and P6, small wild-type oocytes

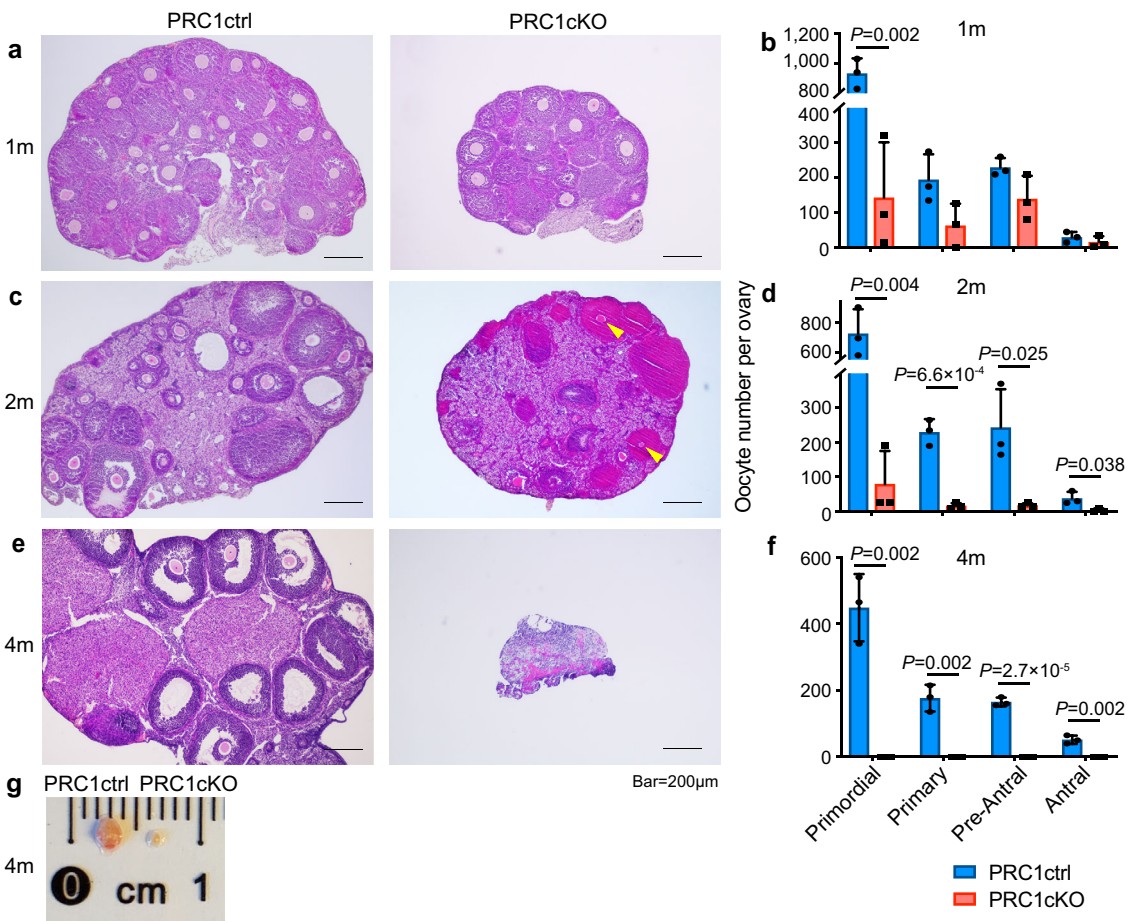

**Fig. 2 Premature ovarian failure in PRC1cKO mice. a**, **c**, **e** Histology of ovarian sections from PRC1ctrl and PRC1cKO females at 1 month, 2 months and 4 months of age, respectively, stained with hematoxylin & eosin. Yellow arrowheads point to atretic follicles. Bars: 200 μm. For each time point, at least three mice of each genotype were used for analysis, and representative images are shown. **b**, **d**, **f** Quantification of the number of different type follicles per ovary from PRC1ctrl and PRC1cKO mice at the age of 1 month, 2 months and, 4 months, respectively. Primordial, Primary, Pre-Antral, and Antral follicles were counted. At least three mice were analyzed for each genotype at each time point. Data are presented as mean values ± SD. Two-tailed unpaired Student's t-test. **g** A photo of ovaries from PRC1ctrl and PRC1cKO females at 4 months of age. Source data are provided as a Source Data file.

differentially expressed 4381 genes, with 2459 showing increased expression and 1922 decreased expression (Supplementary Fig. 3a, Supplementary Data 4). GO analysis revealed that the expression of genes involved in meiosis and reproductive system development is decreased during the POT, whereas genes functioning in the cellular response to oxidative stress are increased, consistent with previous analysis[28] (Supplementary Fig. 3b). Most importantly, genes showing decreased expression during the POT remain highly expressed in PRC1cKO mutant oocytes at P5 (Fig. 3c and Supplementary Fig. 4a). By contrast, genes showing decreased expression during the POT showed significant overlap with genes that were derepressed in PRC1cKO oocytes at P5 (Fig. 3d); overlapping genes included genes involved in blood vessel development, reproduction, kidney morphogenesis, and meiosis (Fig. 3e). Of note, the majority of the 548 overlapping genes belong to Cluster 4 identified in Fig. 1a, which are down-regulated during POT (Supplementary Fig. 4b).

We also tested whether PRC1 suppresses the expression of genes previously identified as being specific for fetal oocytes and MPI[37]. By P5, these genes became repressed in PRC1ctrl oocytes, but remained active in PRC1cKO oocytes (Fig. 3f, g, and Supplementary Fig. 4c). Among 104 MPI genes, 42 overlapped with genes that remain active in P5 PRC1cKO oocytes, and include meiosis initiator genes, *Stra8* and *Meiosin*[38–40]; genes associated with chromosome movement and homologue pairing, such as *Majin*, *Kash5*

(*Ccdc155*)[41–43]; synapsis and recombination related genes, *Msh5*, *Dmc1*, *Hormad1* and *Hormad2*[44–51]; a synaptonemal complex component gene, *Syce3*[52]; a meiosis-specific cohesin component, *Stag3*[53], required for DNA repair and synapsis between homologous chromosomes; a meiosis-specific RPA homologue gene, *Meiob* and its partner protein gene *Spata22*, encoding the single-strand DNA (ssDNA)-binding MEIOB-SPATA22 complex which is required for DNA repair and synapsis[54–57]; and a germ-cell specific gene, *Meioc*, which is required for maintenance of an extended meiotic prophase I[58,59] (Fig. 3h). Taken together, we infer that PRC1 is required to suppress the fetal oocyte program and MPI genes during the POT.

**PRC1-mediated epigenetic programming at POT**. To determine how PRC1 regulates the POT at the chromatin level, we applied CUT&RUN (cleavage under targets and release using nuclease)[60,61] to profile the genomic distributions of (i) PRC1-mediated H2AK119ub, (ii) trimethylation of H3 at lysine 4 (H3K4me3), a representative active mark highly enriched at active promoters near transcription start sites (TSSs) and positively correlated with transcription[62]; and (iii) PRC2-mediated H3K27me3, a repressive mark enriched in inactive promoters and is functionally related to H2AK119ub[8] (Fig. 4 and Supplementary Fig. 5). In wild-type and PRC1ctrl P1 oocytes, the *Hoxa* cluster,

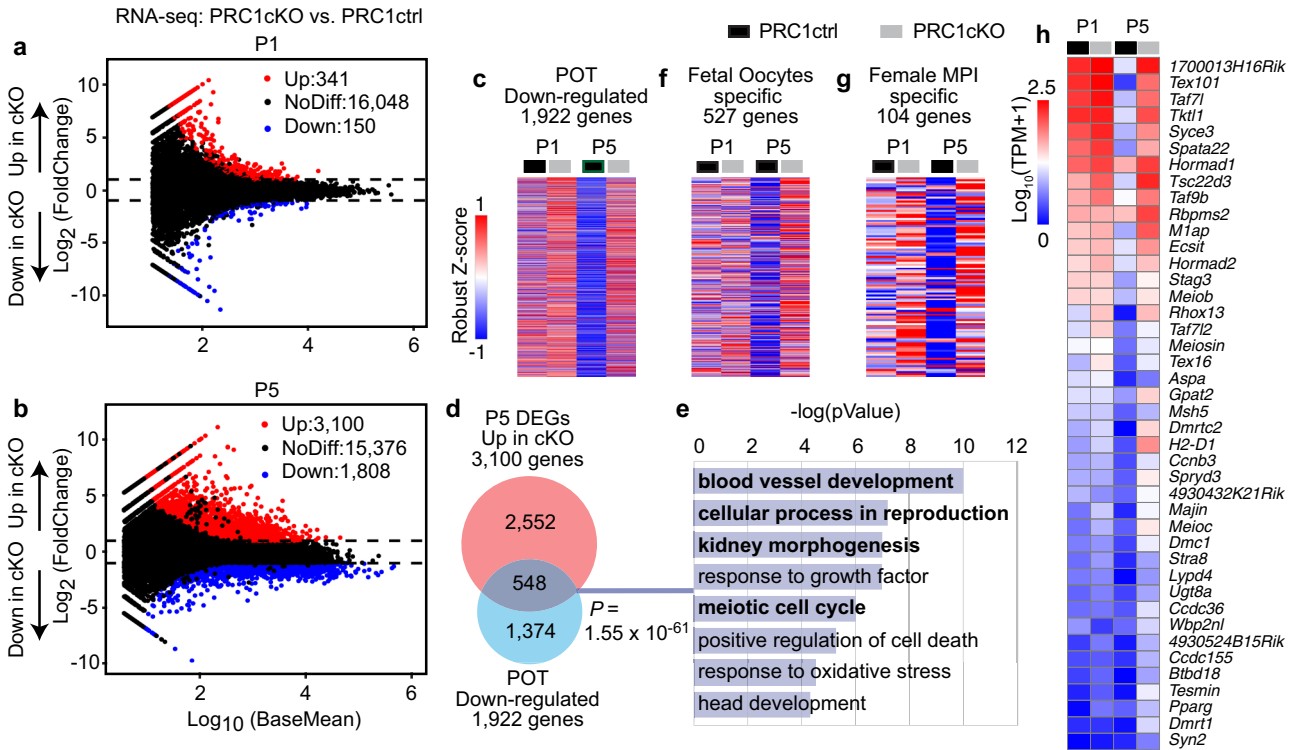

**Fig. 3 PRC1 deficiency causes extensive transcriptional misregulation and defective POT. a, b** Comparison of transcriptomes between PRC1ctrl and PRC1cKO oocytes at P1 and P5, respectively. ~300–500 nongrowing oocytes isolated from P1 or P5 ovaries were pooled as one replicate, and two independent biological replicates were examined for RNA-seq. All the genes with False Discovery Rate (FDR) values are plotted. Differentially expressed genes (DEGs: Log$_2$FoldChange > 1, FDR < 0.05) are colored (red: upregulated in PRC1cKO oocytes; blue: downregulated in PRC1cKO oocytes), and numbers are shown. FDR values are the P$_{adj}$ values generated by DESeq2 by default, in which P values attained by the Wald test were corrected for multiple testing using the Benjamini and Hochberg method. **c, f, g, h** Heatmaps showing expression of POT-down-regulated genes (1922 genes: **c**), fetal oocytes genes (527 genes: **f**), female MPI-specific genes (104 genes: **g**), and 42 overlapped genes between 104 MPI genes and the P5 up-regulated DEGs (**h**) in PRC1ctrl and PRC1cKO oocytes. **d** Overlap between up-regulated DEGs at P5 and POT down-regulated genes. When considering the ratio of POT down-regulated genes to all genes in the genome (1922 genes divided by all 21,169 genes detected in the genome), this association (548 genes divided by 3100 genes) is statistically significant (P = 1.55 × 10$^{-61}$, two-sided hypergeometric test). **e** Key GO enrichments in the overlap detected in **d**. Source data are provided as a Source Data file.

which includes classic Polycomb-targeted developmental genes, was enriched for both H2AK119ub and H3K27me3 (Supplementary Fig. 6a). By contrast, in PRC1cKO P1 oocytes, H3K4me3 was increased on the *Hoxa* cluster, while H3K27me3 remained comparable to controls (Supplementary Fig. 6a). The *Hoxa* cluster is one of the homeobox (*Hox*) gene clusters, and *Hox* genes are involved in various developmental programs, including kidney morphogenesis[63] and blood vessel development[64], which are the representative GO terms for dysregulated genes in PRC1cKO oocytes. These results are consistent with our inference that PRC1 suppresses ectopic expression of inappropriate developmental programs during perinatal oogenesis. Furthermore, we found that PRC1 directly acts on genes that remain highly expressed in PRC1cKO oocytes at P5, e.g., MPI genes, *Mlh1* and *Taf9b*; pro-apoptotic genes, *Bad* (BCL2-associated agonist of cell death) and *Casp7* (Caspase 7); and loss of PRC1's activity at these loci was associated with an increase of H3K4me3 in oocytes at P1 (Fig. 4a, b, Supplementary Fig. 6b). These results suggest that PRC1 counteracts H3K4me3, while the loss of H2AK119ub does not affect the distribution of H3K27me3. Taken together, these results implicate PRC1 in establishing chromatin states that precede, and are likely a prerequisite for, changes in gene expression during the POT.

Consistent with PRCs binding primarily to gene promoters to repress transcription[65–68], all three histone marks were prevalent at gene promoters genome-wide (Fig. 4c, d). Notably, PRC1cKO

mutant oocytes had a global gain of H3K4me3 around TSSs (Fig. 4c, d), particularly on gene Clusters 2-4 (Fig. 4c, Supplementary Fig. 6c), while the overall genomic distribution remained unchanged (Fig. 4c). On the other hand, overall H3K27me3 was modestly reduced, except for gene Cluster 1, which showed a modest increase in mutant oocytes (Fig. 4d, Supplementary Fig. 6c). ChIP-x Enrichment analysis (ChEA) analysis, which detects overlaps with putative binding loci identified in the public next-gen sequencing database, revealed that Cluster 1 genes are common targets of various Polycomb proteins (SUZ12, RNF2, MTF2, JARID2, and EZH2) in embryonic stem cells (ESCs: Supplementary Fig. 7a) and are associated with transcription and developmental processes (Supplementary Fig. 7b). However, as indicated by the increase of H3K4me3 in mutant oocytes, PRC1 regulates a much larger set of genes (9144 genes including Cluster 2-4) that function in a broad range of biological processes including the ubiquitin-proteasome pathway and RNA processing (Cluster 2), DNA metabolic process (Cluster 3), and DNA repair (Cluster 4) (Supplementary Fig. 7b). Consistent with the overall global change, we also observed a significant gain of H3K4me3 and modest reduction of H3K27me3 around TSSs of the 104 MPI genes in PRC1cKO oocytes (Supplementary Fig. 6d), in line with their continued expression at P5. Thus, in addition to canonical Polycomb targets, PRC1 modulates chromatin at a wide range of genes during perinatal oogenesis.

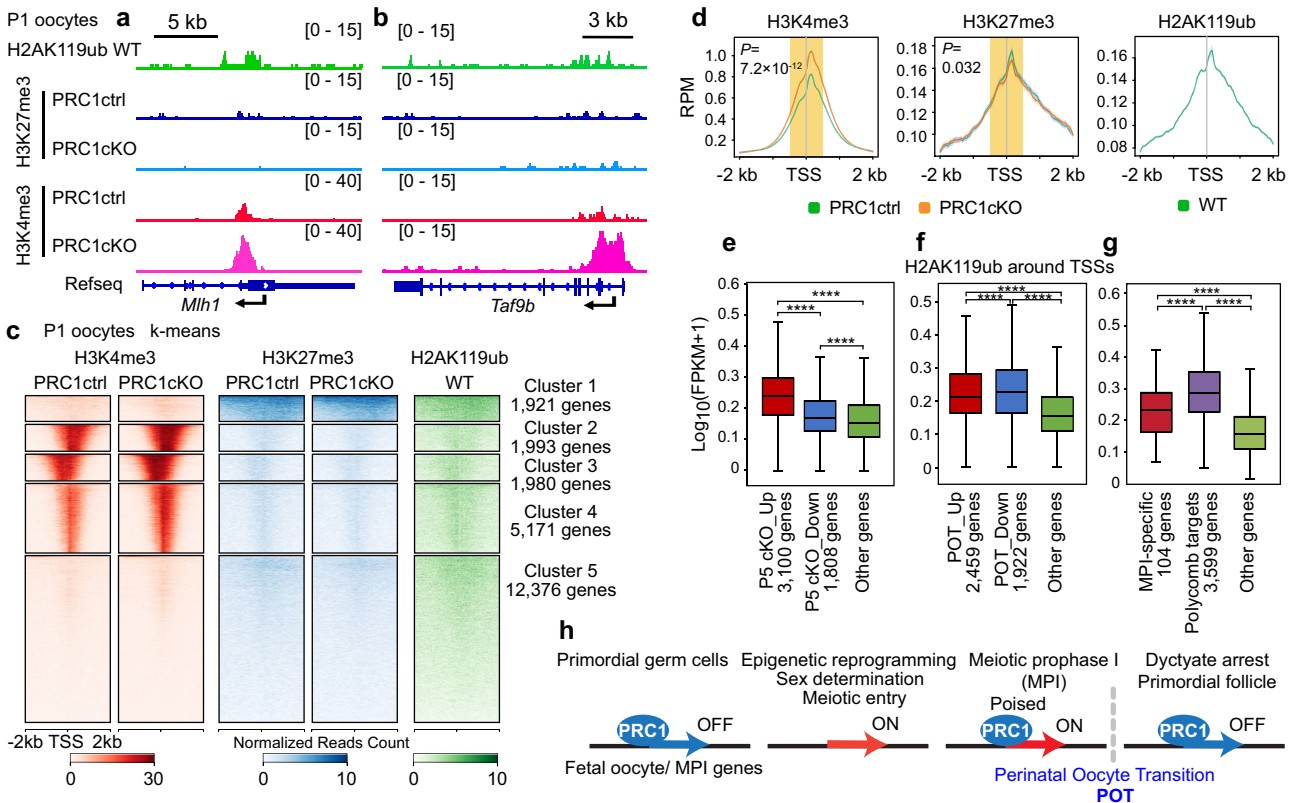

**Fig. 4 Removal of PRC1 affects the global deposition of H3K4me3 in oocytes. a, b** Representative track views showing H2AK119ub, H3K4me3, and H3K27me3 CUT&RUN peaks in P1 oocytes of indicated genotypes. Data ranges are shown in brackets. ~500 nongrowing oocytes isolated from P1 ovaries were pooled as one replicate, and two independent biological replicates were examined for CUT&RUN. **c, d** Heatmaps of k-means clusters and average tag density plots showing H3K4me3, H3K27me3, and H2AK119ub enrichment at promoter region (TSS ± 2 kb) in P1 oocytes. Clusters 2–4 differed in H3K4me3 localization (Cluster 2: H3K4me3 accumulation on 3'-side of transcription start sites (TSSs); Cluster 3: on 5'-side of TSSs; and Cluster 4: on the center of TSSs), but H3K4me3 was highly accumulated in all three clusters. The bars below the heatmaps represent signal intensity, the numbers represent spike-in normalized reads counts. Wilcoxon rank-sum test (two-sided) was performed for read counts in the highlighted area (TSS ± 500 bp). **e, f, g** Box-and-whisker plots showing H2AK119ub enrichment around TSS in different gene groups as indicated. Other genes: $n = 50,343$ (**e**), 50,868 (**f**) and 51,569 (**g**). Central bars represent medians, the boxes encompass 50% of the data points, and the whiskers indicate 90% of the data points. **** $P < 7.0 \times 10^{-7}$ in **e**, **** $P < 2.1 \times 10^{-150}$ in **f**, and **** $P < 4.4 \times 10^{-7}$ in **g**; two-tailed unpaired t-tests. **h** Model of PRC1's function in perinatal oogenesis. Source data are provided as a Source Data file.

To determine whether PRC1 directly regulates the POT, we analyzed H2AK119ub enrichment in P1 oocytes at the TSSs of the genes that subsequently showed dysregulated expression in PRC1cKO at P5. At P1, H2AK119ub was significantly enriched on the TSSs of genes that were ectopically increased at P5 in the PRC1cKO mutant relative to genes whose expression decreased and other genes with unaltered expression (Fig. 4e). Furthermore, H2AK119ub was significantly enriched at the TSSs of the the POT genes when compared to other genes (Fig. 4f), which is also consistent with PRC1 directly regulating changes in gene expression during the POT. To further dissect the role of PRC1 in regulating the meiotic program, we examined the enrichment of H2AK119ub on 104 genes previously defined as being MPI specific[37] and on Polycomb-targeted developmental genes defined in mESCs[69], both of which remain highly expressed at P5 in PRC1cKO mutant oocytes (Supplementary Fig. 4c, d). H2AK119ub was significantly enriched on both the MPI-specific and canonical Polycomb-targeted developmental genes relative to other genes (Fig. 4g). Collectively, these results imply that PRC1 binds the TSSs of target genes to enable the POT.

## Discussion
This study identifies a central role for PRC1 in regulating gene expression during perinatal oogenesis that is critical for establishing the ovarian reserve comprising primordial follicles

(Fig. 4h). The resulting epigenetic landscape and expression program defines the quiescent oocyte state that must be maintained until follicle growth is triggered, which is accompanied by another dramatic wave of epigenetic programming including the re-methylation of DNA[21]. Following meiotic entry, driven by the release of MPI genes from PRC1-mediated silencing[16–18], MPI genes are again targeted by PRC1 for suppression during the POT. Thus, PRC1 regulates both the entry and exit from the MPI program in the female germline, with PRC1-dependent chromatin programming preceding the dramatic changes in gene expression that occur during perinatal oogenesis. Mechanistically, in P1 oocytes, PRC1 counteracts H3K4me3 at the TSSs of a large number of genes expressed during fetal oogenesis and these genes are then suppressed after the POT (Fig. 4h). How PRC1 function is regulated during this critical window of oogenesis is the subject of ongoing investigation.

During oogenesis in PRC1cKO mice, extensive cell death takes place between P1 and P5, leading to a small ovarian reserve. During this developmental window, we find massive transcriptional dysregulation in PRC1cKO oocytes. Although we are not able to specify a cause of oocyte death, we observed that genes differentially expressed in P5 PRC1cKO oocytes are enriched with genes associated with "positive regulation of cell death" and "autophagy" (Supplementary Fig. 2), including pro-apoptotic genes, *Bad* and *Casp7*, and PRC1 directly acts on *Bad* and *Casp7*

gene loci (Supplementary Fig. 6b). Therefore, it is possible that PRC1 directly regulates apoptosis by regulating expression of BAD and CASP7. Alternatively, it is also possible that continued expression of MPI genes could be toxic and induce oocyte apoptosis/autophagy.

At P5, a small number of PRC1cKO oocytes remain despite the highly efficient *Ddx4*-Cre-mediated deletion of PRC1. One possible reason why the ovarian reserve is still present at P5 could be attributed to the heterogeneity in ovarian reserve formation. There are two regions of primordial follicle formation in the mouse ovary. Assembly of primordial follicles takes place from E17.5 to P5 in the medulla and cortex. The medullary follicles start to grow as soon as they are formed, whereas the cortical primordial follicles mature gradually over the reproductive life-span of the animal[70]. Because *Ddx4*-Cre expression initiates in fetal germ cells from E15, it is possible that a certain number of primordial follicles have formed before PRC1 is completely inactivated. Of note, we found a gradual decrease in the number of PRC1cKO primordial follicles with age, suggesting a role of PRC1 in maintaining the dictyate stage.

In summary, our findings establish a critical role for PRC1 in establishing a stable ovarian reserve. Thus, deficiencies in PRC1 function may underlie specific instances of premature ovarian failure and infertility in human females.

## Methods

**Animals**. Generation conditionally deficient *Rnf2* mice on a *Ring1*$^{-/-}$ background was performed as described previously[35]. Briefly, PRC1cKO mice *Ring1*$^{-/-}$; *Ring1*$^{F/-}$; *Ddx4*-Cre were generated from *Ring1*$^{-/-}$; *Ring1*$^{F/F}$ females crossed with *Ring1*$^{-/-}$; *Ring1*$^{F/+}$; *Ddx4*-Cre males, and PRC1ctrl mice used in experiments were *Ring1*$^{-/-}$; *Rnf2*$^{F/+}$; *Ddx4*-Cre littermate females. Mice were maintained on a mixed genetic background of FVB and C57BL/6 J. Generation of mutant *Ring1* (*Ring1*$^{-/-}$) and *Rnf2* floxed alleles (*Ring1*$^{F/F}$) were reported previously[34,71]. *Ring1*$^{-/-}$ and *Ring1*$^{F/F}$ mice were obtained from Dr. Haruhiko Koseki. *Ddx4*-Cre transgenic mice were purchased from the Jackson Laboratory[32]. *Stella*-GFP transgenic mice were obtained from Dr. M. Azim Surani[72]. For each experiment, a minimum of three independent mice was analyzed. Mice were maintained on a 12:12 light:dark cycle in a temperature and humidity-controlled vivarium (22 ± 2 °C; 40–50% humidity) with free access to food and water in the pathogen-free animal care facility. Mice were and used according to the guidelines of the Institutional Animal Care and Use Committee (IACUC: protocol no. IACUC2018-0040 and 21931) at Cincinnati Children's Hospital Medical Center and the University of California, Davis.

**Antibodies**. A list of antibodies used in this study is in the Supplemental Material (Supplementary Table 1).

**Oocyte collection**. The P1 or P5 female pups were collected, and ovaries were harvested by carefully removing oviducts and ovarian bursa in PBS. Each pair of ovaries were further digested in 200 µl TrypLE™ Express Enzyme (1X) (Gibco, 12604013) supplemented with 0.25% Collagenase Type 1 (Worthington, CLS-1) and 0.01% DNase I (Sigma, D5025) and incubated at 37°C for 15 min with gentle agitation. After incubation, the ovaries were dissociated by gentle pipetting using the Fisherbrand™ Premium Plus MultiFlex Gel-Loading Tips until no visible tissue pieces. 2 ml DMEM/F-12 medium (Gibco, 11330107) supplemented with 10% FBS (HyClone, SH30396.03) were then added to the suspension to stop enzyme reaction. Cell suspension was filtered with 40 µm cell strainer (Falcon, 352340) and seeded onto 35 mm tissue culture dish (Falcon, 353001). The cells were allowed to settle down for 15 min at 37°C; 5% CO$_2$ before being transferred under the microscope (Nikon, SMZ1270). Based on morphology and diameter, non-growing oocytes (Supplementary Fig. 2a, yellow arrowheads and arrows point to typical single non-growing oocytes among ovarian cells at P1 and P5, respectively) were specifically picked up, washed, and transferred into the downstream buffer by mouth pipette.

**Histology and Immunostaining**. For the preparation of ovarian paraffin blocks, ovaries were fixed with 4% paraformaldehyde (PFA) overnight at 4 °C. Ovaries were dehydrated and embedded in paraffin. For histological analysis, 5-µm-thick paraffin sections were deparaffinized and stained with hematoxylin and eosin. For immunostaining, ovarian paraffin sections were deparaffinized and autoclaved in target retrieval solution (DAKO) for 10 min at 121 °C. Sections were blocked with Blocking One Histo (Nacalai) for 1 h at room temperature and then incubated with primary antibodies (anti-H2AK119ub, anti-DDX4, and anti-SYCP3 at 1:200 dilution) overnight at 4 °C. Localization of the primary antibody was performed by

incubation of the sections with the corresponding secondary antibodies (Donkey Anti-Mouse IgG (H + L) Alexa Fluor 488, A-21202; Donkey Anti-Rabbit IgG (H + L) Alexa Fluor 555, A-31572; Donkey Anti-Rabbit IgG (H + L) Alexa Fluor 488, A-21206; Donkey Anti-Mouse IgG (H + L) Alexa Fluor 555, A-31570; Invitrogen) at 1:500 dilution for 1 h at room temperature. Finally, sections were counterstained with DAPI and mounted using 20 mL undiluted ProLong Gold Antifade Mountant (ThermoFisher Scientific, P36930). Images were obtained by confocal laser scanning microscope (A1R, Nikon) and processed with NIS-Elements (Nikon).

**Quantification of ovarian follicles**. Quantification of ovarian follicles was performed as previously described[73]. Briefly, to count the numbers of follicles, paraffin-embedded ovaries were serially sectioned at 5-µm thickness, and all sections were mounted on slides. Then these sections were stained with hematoxylin and eosin for morphological analysis. Ovarian follicles at different developmental stages, including primordial, primary (type 3 and type 4), pre-antral (type 5), and antral follicles (type 6 and type 7), were counted in every fifth section of the collected sections from one ovary, based on the well-accepted standards established by Pederson and Peters[74]. In each section, only those follicles in which the nucleus of the oocyte was clearly visible were scored and the cumulative follicle counts were multiplied by a correction factor of 5 to represent the estimated number of total follicles in an ovary.

**Meiotic chromosome spreads and immunofluorescence**. Chromosome spreads of oocytes from neonatal ovaries were prepared as described[75]. Briefly, neonatal ovaries were harvested and incubated in hypotonic extraction buffer [HEB: 30 mM Tris base, 17 mM trisodium citrate, 5 mM ethylenediaminetetraacetic acid (EDTA), 50 mM sucrose, 5 mM dithiothreitol (DTT), 1× cOmplete Protease Inhibitor Cocktail (Sigma, 11836145001), 1× phosphatase inhibitor cocktail 2 (Sigma, P5726-5ML), pH 8.2] on ice for 30 min with gentle stirring every 10 min. Then, a suspension of oocytes was generated by pipetting an ovary using Fisherbrand™ Premium Plus MultiFlex Gel-Loading Tips in 30 µL of sucrose (100 mM). 30 µL of the suspension was applied to positively charged slides (Probe On Plus: Thermo Fisher Scientific, 22-230-900); before application of the suspension, the slides had been incubating in chilled fixation solution (2% paraformaldehyde, 0.1% Triton X-100, 0.02% sodium monododecyl sulfate, adjusted to pH 9.2 with sodium borate buffer) for a minimum of 2 min. The slides were placed in "humid chambers" at RT for a minimum of 1 h (maximum overnight). Then, the slides were washed twice in 0.4% Photo-Flo 200 (Kodak, 146–4510), 2 min per wash. Slides were dried completely at RT before staining or storage in slide boxes at −80 °C.

For immunostaining experiments, oocyte chromosome spreads were washed by PBS containing 0.1% Tween 20 (PBST) for 5 min before blocking in blocking buffer (PBST containing 1% BSA) for an additional 30 min. Primary and secondary antibodies were diluted in PBST containing 0.15% BSA. Primary antibodies were used at the following dilutions: rabbit anti-SYCP3 (Novus, NB300-232), 1/500; mouse anti-γH2AX (Millipore, 05-636), 1/5000; mouse anti-SYCP3 (Abcam, ab97672), 1/5000; mouse anti-SYCP3 conjugated with Alexa 488 fluorophore (Abcam, ab205846), 1/500; mouse anti-γH2AX conjugated to Alexa 647 fluorophore (Millipore, 05-636-AF647), 1/500. The following steps are the same as described previously for immunostaining of paraffin sections.

**RNA-seq library generation and sequencing**. RNA-seq libraries of PRC1ctrl and cKO oocytes were prepared as follows. ~300-500 non-growing oocytes isolated from P1 or P5 ovaries were pooled as one replicate, and two independent biological replicates were used for RNA-seq library generation. Total RNA was extracted using the RNeasy plus Micro Kit (QIAGEN, Cat # 74034) according to the manufacturer's instructions. Library preparation was performed with NEBNext® Single Cell/Low Input RNA Library Prep Kit for Illumina® (NEB, E6420S) according to the manufacturer's instruction. Prepared RNA-seq libraries were sequenced on the HiSeq 4000 system (Illumina) with paired-ended 150-bp reads.

**CUT&RUN library generation and sequencing**. CUT&RUN libraries of PRC1ctrl, PRC1cKO, and *Stella*-GFP oocytes were conducted as previously described[61]. ~500 non-growing oocytes isolated from P1 ovaries were pooled as one replicate, and two independent biological replicates were used for CUT&RUN library generation. The antibodies used were rabbit anti-H2AK119ub1 (1/100; Cell Signaling Technology; 8240), rabbit anti-H3K4me3 (1/100; Cell Signaling Technology; 9751), and rabbit anti-H3K27me3 antibody (1/100; Cell Signaling Technology; 9733). Libraries were constructed using NEBNext® Ultra II DNA Library Prep Kit for Illumina (NEB, E7645S). Prepared CUT&RUN libraries were sequenced on the NovaSeq 6000 system (Illumina) with paired-ended 100-bp or 50-bp reads.

**RNA-seq data processing**. Raw RNA-seq reads after trimming by Trim-galore (https://github.com/FelixKrueger/TrimGalore) (version 0.6.6) with the parameter '--paired' were aligned to the mouse (GRCm38/mm10) genome using HISAT2[76] (version 2.2.1) with default parameters. All unmapped reads, non-uniquely mapped reads, reads with low mapping quality (MAPQ < 30) were filtered out by samtools[77] (version 1.9) with the parameter '-q 30' before being subjected to downstream analyses.

To identify differentially expressed genes in RNA-seq, raw read counts for each gene were generated using htseq-count function, part of the HTSeq package[78] (version 1.6.0) based on the mouse gene annotation (gencode.vM24.annotation.gtf, GRCm38/mm10). DESeq2[79] (version 1.28.1) was used for differential gene expression analyses with cutoffs FoldChange > 2 and FDR values (P adjusted: $P_{adj}$ values) < 0.05. In DESeq2, P values attained by the Wald test were corrected for multiple testing using the Benjamini and Hochberg method by default (FDR values). FDR values were used to determine significantly dysregulated genes. The TPM values of each gene were generated using RSEM[80] (version 1.3.3) for comparative expression analyses and computing the Pearson correlation coefficient between biological replicates.

To perform GO analyses, we used the online functional annotation clustering tool Metascape[81] (http://metascape.org). Further bioinformatics analyses were visualized as heatmaps using Morpheus (https://software.broadinstitute.org/morpheus/). The TPM values of each gene were first transformed by Robust Z-Score Method, also known as the Median Absolute Deviation method, or log transformed, then used as input for heatmap plotting.

**CUT&RUN data processing**. For CUT&RUN data processing, we basically followed the online tutorial posted by Henikoff Lab (https://protocols.io/view/cut-amp-tag-data-processing-and-analysis-tutorial-bjk2kkye.html).

Briefly, raw paired-end reads after trimming by Trim-galore (https://github.com/FelixKrueger/TrimGalore) (version 0.6.6) were aligned mouse (GRCm38/mm10) genome using Bowtie2[82] (version 2.4.2) with options: --end-to-end --very-sensitive --no-mixed --no-discordant --phred33 -I 10 -X 700. E. coli DNA carried along with bacterially-produced pAG-MNase protein brings along a fixed amount of E. coli DNA and can be used as spike-in DNA. For mapping E.coli spike-in fragments, we also use the --no-overlap --no-dovetail options to avoid cross-mapping. PCR duplicates were removed from low-cell-number data using the 'MarkDuplicates' command in Picard tools (version 2.23.8) (https://broadinstitute.github.io/picard/). To compare replicates, Pearson correlation coefficients were calculated and plotted by multiBamSummary bins and plot correlation from deepTools[83] (version 3.5.0). Biological replicates were pooled for the visualization and other analyses after validation of the reproducibility (Pearson correlation coefficient > 0.85). Spike-in normalization was implemented using the exogenous scaling factor computed from the E.coli mapping files (the formula used for calculation of scaling factors = 1/ (spike-in reads/100,000)). For visualization of CUT&RUN using the Integrative Genomics Viewer[84] (version 2.5.3), spike-in normalized genome coverage tracks were generated using genomecov function of bedtools[85] (version 2.29.2) with '-bg -scale $scale_factor' parameter. SEACR[86] (https://seacr.fredhutch.org/) was used for peak calling. The program ngs.plot[87] (version 2.63) was used to draw tag density plots and heatmaps for read enrichments. The FPKM values of each gene's promoter region (TSS ± 2 kb) were generated using Bamscale package[88] (version 0.0.5) for comparative expression analyses.

Gene Ontology Biological Process (GO-BP) term enrichment and ChIP-x Enrichment Analysis (ChEA) using the gene sets clustered from the CUT&RUN results was performed using Enricher website (https://maayanlab.cloud/Enrichr/)[89,90]. Dot plot was produced using the R package ggplot2 (3.3.6).

**Statistics**. Statistical methods and P values for each plot are listed in the figure legends and/or in the Methods. Statistical significances for pairwise comparisons were determined using two-tailed Wilcoxon rank-sum test and unpaired t-tests. Next-generation sequencing data (RNA-seq, CUT&RUN) were based on two independent replicates. For all experiments, no statistical methods were used to predetermine sample size. Experiments were not randomized, and investigators were not blinded to allocation during experiments and outcome assessments.

**Reporting summary**. Further information on research design is available in the Nature Research Reporting Summary linked to this article.

## Data availability

RNA-seq data and CUT&RUN data reported in this study were deposited to the Gene Expression Omnibus (accession no. GSE184208). Source data are provided with this paper.

## Code availability

Source code for all software and tools used in this study with documentation, examples and additional information, is available at URLs listed above.

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

## Acknowledgements

We thank Azusa Inoue for critical reading of the manuscript; Artem Barski for sharing reagents; M. Azim Surani for sharing *Stella*-GFP transgenic mice, Wei Xie for sharing the CUT&RUN protocol and the Polycomb targets gene list; Yuki Horisawa-Takada and Kei-Ichiro Ishiguro for discussion of PRC1's function in gametogenesis. Funding sources: National Institutes of Health grants R01GM122776 and R35GM141085 to S.H.N.

## Author contributions

M.H. and S.H.N. designed the study. M.H., Y.H.Y., H.A., and S.M. performed experiments. M.H., Y.M. A.S., and S.H.N. designed and interpreted the computational analyses. M.V. and H.K provided key materials. M.H., N.H., R.M.S., and S.H.N. interpreted the results and wrote the manuscript with critical feedback from all other authors. R.M.S. and S.H.N. supervised the project.

## Competing interests

The authors declare no competing interests.

**Additional information**

