## [Peer Review File · Nature Communications]

PRC1-mediated epigenetic programming is required to generate the ovarian reserveREVIEWER COMMENTS

Reviewer #1 (Remarks to the Author):

In this manuscript, Hu et al identify a transcriptional profile of a large set of differentially expressed genes between PN1 and PN3 as oocytes transition from meiotic prophase I to dictyate arrest, which they term the perinatal oocyte transition (POT). As the downregulation of the meiotic gene program during the POT and subsequent requirement for reestablishment of pluripotency during this period appears to be a reversal of the earlier transition from pluripotent primordial germ cell to meiosis, the authors hypothesized that the polycomb repressive complex 1 (PRC1) is similarly required. They generate a germ cell conditional knockout of PRC1 with a RNF2 floxed allele using Ddx4-Cre on a background of Ring1 deficiency, given its partial redundancy. Phenotypically, oocytes progress through meiosis I but start to deplete at PN5, with all stages of follicles decreasing thereafter until 4 months, when no oocytes remain. By RNAseq they show that POT-associated genes remain high in PRC1-cKO oocytes, including those associated with meiosis and the differentiation of multiple tissues. They find that in mutant oocytes, the active chromatin mark H3K4m3 is increased at developmental-specific genes such as the Hox cluster as well as meiotic gene loci, and that these new sites of H3K4m3 decoration correspond with the genes downregulated during the POT. These results inform the authors' model that PRC1 directly silences the meiotic gene program between PN1 and PN3 and maintains the silence of other differentiation programs as oocytes move from meiotic prophase to early follicles by directly binding to TSSs of these genes.

Overall, this study identifies a new role for PRC1 in neonatal oocyte in the silencing of the meiotic program as well as inappropriate developmental programs, which is critical for establishing the ovarian reserve of oocytes for lifelong fertility. This dual function of PRC1 in regulating both the entry and exit from meiotic prophase I opens up new mechanistic questions.

This study is beautifully and rigorously executed. My minor comments follow:

1. For completeness, please describe the dynamic of expression of Ring1 and Rnf2f in germ cells during the period of study.
2. Fig 1d: what is the orientation of the ovary? It seems to be different in the IF images compared to the H&E, where hylum is up in most, except mutant P5. Please represent consistently.
3. Fig 1e-f are confusing as f seems to be showing the trend across time whereas e depicts individual datapoints. Please blow up the scale below 2000 on e so that differences can be appreciated.
4. It should be stated in the text that the oocyte counts in Figure 1e-f are estimates from representative sections rather than direct counts.
5. In Fig 2 the numbers of growing follicles are difficult to see in the graphs and it is recommended that they should be shown on different scales with a break on the axis.
6. The numbers shown in Figure 2 imply that even growing follicles are increasingly dying in the PRC1 cKO, in addition to primordial follicles. This point can be addressed by comparing the frequency of antral to primary for mutant vs WT at each age. It may be consistent with the hypothesis that even those mutant oocytes which manage to begin growth are unable to complete gametogenesis due to their inability to suppress meiotic or developmentally inappropriate genes.
7. In fig 3, how does the transcriptional profile of P5 mutant oocytes compare to P1 WT?
8. It is a very small issue, but the use of red to denote both gene upregulation and PRC1 cKO in Figure 3 is confusing, and I would suggest selecting different sets of colors for each.
9. On p. 5, the statement "These data imply that PRC1 is required immediately after MPI for oocytes to transition to dictyate arrest" seems misleading, as in Fig. S1C it is shown by IF and quantified that oocytes in PRC1 cKO reach dictyate at equivalent time and frequency compared to WT. It seems that there is a disconnect between the progression of the nucleus through MPI and the transcriptional state in the cytoplasm. Could the authors please make this distinction in their interpretation.

Reviewer #2 (Remarks to the Author):

Summary of submission:

In this study, the authors report that PRC1 is important for the progression from meiotic prophase I and into a quiescent state (dictyate arrest), in addition to suppressing other developmental programmes. They report an expression profile of the meiotic prophase I expression pattern in the PRC1 conditional knockout as well as a loss of oocytes at the P5 stage, when the ovarian reserve is established.

General comments:

This is a well conducted study and the findings are significant and interesting to the broader community. They support the conclusion that the ovarian reserve is reduced in the PRC1 cKO and that, by inference, PRC1 has an important role therein.

I have some questions that should be addressed prior to publication:

1. It is not made clear what causes oocyte death in the PRC1 cKO. Continued expression of MPIs which trigger apoptosis/autophagy?
2. Would the authors please comment on the variability in the perinatal oocytes – does Ddx4 deplete all oocytes of PRC1 or is PRC1 expression maintained in a subset, which would explain why the ovarian reserve at P5 isn't 0? However, it wouldn't necessarily explain why there's a decline over time in the cKO; would that perhaps indicate a role of PRC1 in the continued maintenance of the dictyate stage?
3. Bioinformatic analyses and integration.
These are very challenging experiments and they have a high degree of reproducibility, especially the CUT&RUN. There are some clarifications that would be helpful.
 - i. I think a deeper analysis of the 104 MPI genes and/or other genes that might explain the potential causes that result in apoptosis (see point 1) would strengthen the study.
 - ii. Figure 1a – the heatmap are shown and the clusters are defined. However, it wasn't quite clear what a robust z-score is and how this relates to a false discovery rate. It would be helpful if this aspect of the higher level analysis wasn't well described in the Methods.
 - iii. The central hypothesis is that altered H2K119ub is correlated with elevated H3K4me3 and increased expression of meiotic cell cycle (MPI) genes in the PRC1 cKO leading to oocyte atresia. I might have missed this, but what is the overall correlation between loss of H2AK119ub and gain of H3K4me3 across DEGs? Especially in the meiotic cell cycle genes? I think that the overlap of H2AK119ub is shown for the 104 MPIs in Figure 4f, but not what happens to the chromatin marks and expression of the 104 genes in the cKO?
 - iv. Figure 3 – it would be really good with an integration into the findings from figure 1; e.g, which clusters to the differentially expressed genes belong to? A heatmap of the same clusters as in Figure 1a, would also be good to assess what the general, overall transcriptional effect is in the cKO between P1 and P5 in comparison to the changes occurring in the wild type.
 - v. Some of the comparisons of chromatin modification use profiles from mESCs. This is understandable as these analyses are challenging. However, I think the authors would need to justify why mESC profiles are appropriate for the chromatin marks. Or at least discuss uncertainties, since this is a different stage of development and many of the marks show large changes during gametogenesis and preimplantation development.

Other comments:

A short introduction to PRC1 would be good.

Line 101: reached to the -> reached the.

Figure 1 – legend. Spell out abbreviations again in legend (POT, PPT). Please state how many oocytes were included in the RNA-seq analyses.

Figure 3a – it would be really good if the authors added a brief explanation of the FDR calculations in the legend.

Figure 4c – what is the interpretation of the H3K4me3 distribution, which has shifted in cluster 2 compared to the other clusters? Define bars (e.g. 0 to 30) below the plots.

Extended data figure 2e – it's hard to read the list of meiotic cell cycle genes. It might be better to move this list to a supplemental table?

Reviewer #3 (Remarks to the Author):

The manuscript by Hu et al. demonstrated that PRC1-mediated histone modification, most likely H2AK119ub, is required for exit of the gene expression program underlying meiotic prophase I (MPI) in perinatal oocytes. The authors first identify a large gene expression transition, perinatal oocyte transition (POT), between postnatal days 1 and 3. As its general repressive effect on gene expression, the author hypothesize that PRC1 play a role on the POT. In PRC1-deficient oocytes, aberrant gene expression, many of which are upregulated, was observed in P5 oocytes, and a part of these gene loci show an elevated level of H3K4me3. Importantly, genes repressed by PRC1 at POT contain meiotic genes involved in MPI. Thus, PRC1 plays a critical role on POT by repressing meiotic genes.

Using a mouse model, the authors demonstrate the role of PRC on POT, which has novelty for a better understanding gene expression regulation during early oogenesis. Importantly, genes repressed by PRC1 at POT are the genes repressed in PGCs until MPI entry, illustrating a temporal regulation of meiotic genes by PRC1. The results are clearly shown, though several points to be considered further (see below). This reviewer in principle support to publish this paper in Nature Communications.

Comments:

1. Looking at gene expression dynamics between E18.5 to P6, genes repressed by PRC1 at POT should belong to the cluster 4, as other clusters do not show clear difference in gene expression upon POT. Would it be true? For example, do 548 genes shown in Figure 3d belong to the cluster 4? Showing such information would be more informative.

2. I cannot quite understand why the author perform CUT&RUN analysis using only P1 rather than using also P3 or P5, where difference in gene expression is clear. Given repression by PRC1, such analysis would reinforce the author's claim.

3. Related to the comment 2, it is not entirely clear whether the repressive effect of PRC1 is mediated H2AK119ub, as Figure 4c, which show dim signals, is the only evidence supporting the claim. Would it be possible to clear this point? Or, if impossible, the author should discuss this point more intensively.

4. L168: Furthermore, we found that PRC1 directly binds to genes that remain highly expressed in PRC1cKO oocytes at P5, e.g., Mlh1 and Taf9b; and loss of PRC1 binding at these loci was associated with an increase of H3K4me3 in oocytes at P1 (Fig. 4a and 4b). What is the evidence demonstrating this? Figure 4a and b show histone marks but not PRC1.

5. Figure 1e and f are redundant. Delete f, or explain why both are shown.

RESPONSE TO REVIEWERS

GENERAL COMMENTS FOR ALL REVIEWERS

We thank the Reviewers for their careful consideration of our manuscript (NCOMMS-22-10083-T) and their helpful comments, which led to revisions that we believe significantly improved the manuscript. All three Reviewers found our study significant and well-conducted and were positive about the study. To address the points raised by the Reviewers, we have modified Figures 1, 2, 3, and 4; included new analyses in Supplementary Figures 2, 4, and 6; and revised the text as suggested. Below, our specific comments to each Reviewer follow. Please note: *Reviewer comments are in italics* and **our responses are in bold**. The Reviewers' comments have not been edited.

Reviewer #1 (Remarks to the Author)

In this manuscript, Hu et al identify a transcriptional profile of a large set of differentially expressed genes between PN1 and PN3 as oocytes transition from meiotic prophase I to dictyate arrest, which they term the perinatal oocyte transition (POT). As the downregulation of the meiotic gene program during the POT and subsequent requirement for reestablishment of pluripotency during this period appears to be a reversal of the earlier transition from pluripotent primordial germ cell to meiosis, the authors hypothesized that the polycomb repressive complex 1 (PRC1) is similarly required. They generate a germ cell conditional knockout of PRC1 with a RNF2 floxed allele using Ddx4-Cre on a background of Ring1 deficiency, given its partial redundancy, Phenotypically, oocytes progress through meiosis I but start to deplete at PN5, with all stages of follicles decreasing thereafter until 4 months, when no oocytes remain. By RNAseq they show that POT-associated genes remain high in PRC1-cKO oocytes, including those associated with meiosis and the differentiation of multiple tissues. They find that in mutant oocytes, the active chromatin mark H3K4m3 is increased at developmental-specific genes such as the Hox cluster as well as meiotic gene loci, and that these new sites of H3K4me3 decoration correspond with the genes downregulated during the POT. These results inform the authors' model that PRC1 directly silences the meiotic gene program between PN1 and PN3 and maintains the silence of other differentiation programs as oocytes move from meiotic prophase to early follicles by directly binding to TSSs of these genes.

Overall, this study identifies a new role for PRC1 in neonatal oocyte in the silencing of the meiotic program as well as inappropriate developmental programs, which is critical for establishing the ovarian reserve of oocytes for lifelong fertility. This dual function of PRC1 in regulating both the entry and exit from meiotic prophase I opens up new mechanistic questions.

This study is beautifully and rigorously executed. My minor comments follow:

1. For completeness, please describe the dynamic of expression of Ring1 and Rnf2 in germ cells during the period of study.

We now include a heatmap showing the expression dynamics of *Ring1* and *Rnf2* in Fig. 1a and added a corresponding description in the main text (Line 90).

2. Fig 1d: what is the orientation of the ovary? It seems to be different in the IF images compared to the H&E, where hylum is up in most, except mutant P5. Please represent consistently.

Thank you for catching this point. We adjusted the orientation of the ovaries and presented them consistently in Fig. 1d.

3. Fig 1e-f are confusing as f seems to be showing the trend across time whereas e depicts individual datapoints. Please blow up the scale below 2000 on e so that differences can be

appreciated.

In these panels, we intended to show the distribution of individual data points using the scatter plot in Fig. 1e and show the trend across time points using the line chart in Fig. 1f. In response to the reviewer's suggestion, we updated the plot in Fig. 1e and blew up the scale below 2000.

4. It should be stated in the text that the oocyte counts in Figure 1e-f are estimates from representative sections rather than direct counts.

We reworded the main text and figure legend to clarify that the oocyte counts in Figs. 1e-f are estimates from representative sections (Line 100).

5. In Fig 2 the numbers of growing follicles are difficult to see in the graphs and it is recommended that they should be shown on different scales with a break on the axis.

We updated these panels as suggested.

6. The numbers shown in Figure 2 imply that even growing follicles are increasingly dying in the PRC1 cKO, in addition to primordial follicles. This point can be addressed by comparing the frequency of antral to primary for mutant vs WT at each age. It may be consistent with the hypothesis that even those mutant oocytes which manage to begin growth are unable to complete gametogenesis due to their inability to suppress meiotic or developmentally inappropriate genes.

In response to the Reviewer's concern, we performed the following two analyses using our data sets to evaluate the increased death of growing oocytes in PRC1cKO ovaries. As this Reviewer anticipated, these analyses confirmed this hypothesis.

- 1) We found that the estimated numbers of primary follicles per ovary continue to decrease from 1 month (1m) to 4m in PRC1cKO ovaries, while those present in control ovaries (PRC1ctrl) remain largely unchanged from 1m to 4m (Figure R1a, below).
- 2) We examined the estimated rates of development from primary follicles of a previous time point to the antral follicle of the next time point by comparing antral and primary follicle numbers. This rate decreases with age in PRC1cKO ovaries, confirming the increased death of growing oocytes in PRC1cKO ovaries (Figure R1b, below).

Figure R1: Increased death of growing oocytes in PRC1cKO ovaries.

(a) Estimated numbers of primary follicles per ovary.

(b) Estimated rates of development from primary follicles (P) of a previous time point to antral follicles (A) of a following time point.

7. In fig 3, how does the transcriptional profile of P5 mutant oocytes compare to P1 WT?

We believe that this question might be a misunderstanding. In Fig. 3, we did not compare the transcription profile of P5 mutant oocytes to P1 WT. Fig. 3 is based on pairwise comparisons between ctrl and cKO at two time points (P1 and P5).

8. It is a very small issue, but the use of red to denote both gene upregulation and PRC1 cKO in Figure 3 is confusing, and I would suggest selecting different sets of colors for each.

We now use Black/Grey to show genotypes in Fig. 3.

9. On p. 5, the statement “These data imply that PRC1 is required immediately after MPI for oocytes to transition to dictyate arrest” seems misleading, as in Fig. S1C it is shown by IF and quantified that oocytes in PRC1 cKO reach dictyate at equivalent time and frequency compared to WT. It seems that there is a disconnect between the progression of the nucleus through MPI and the transcriptional state in the cytoplasm. Could the authors please make this distinction in their interpretation.

We apologize for this confusion and agree with the Reviewer’s point. These data imply that PRC1 is required for the transcriptional transition from MPI to dictyate arrest. We reworded this statement in text (Lines 146-148).

Reviewer #2 (Remarks to the Author):

Summary of submission:

In this study, the authors report that PRC1 is important for the progression from meiotic prophase I and into a quiescent state (dictyate arrest), in addition to suppressing other developmental programmes. They report an expression profile of the meiotic prophase I expression pattern in the PRC1 conditional knockout as well as a loss of oocytes at the P5 stage, when the ovarian reserve is established.

General comments:

This is a well conducted study and the findings are significant and interesting to the broader community. They support the conclusion that the ovarian reserve is reduced in the PRC1 cKO and that, by inference, PRC1 has an important role therein.

I have some questions that should be addressed prior to publication:

1. It is not made clear what causes oocyte death in the PRC1 cKO. Continued expression of MPIs which trigger apoptosis/autophagy?

Because we found massive transcriptional dysregulation in PRC1cKO oocytes, we were not able to specify a cause of oocyte death. However, we observed that genes differentially expressed in P5 PRC1cKO oocytes are enriched with genes associated with “positive regulation of cell death” and “autophagy” (Supplementary Fig. 2). These genes include pro-apoptotic genes, *Bad* (BCL2-associated agonist of cell death) and *Casp7* (Caspase 7), and PRC1 directly binds and regulates *Bad* and *Casp7* gene loci (New Supplementary Fig. 6b). Therefore, it is possible that PRC1 directly regulates apoptosis by regulating expression of BAD and CASP7. As the Reviewer pointed out, it is also possible that continued expression of MPI genes could be toxic and induce oocyte apoptosis/autophagy. We do not have any data that address this possibility, noting that we now raise this point in the Discussion (Lines 256-265).

*2. Would the authors please comment on the variability in the perinatal oocytes – does *Ddx4* deplete all oocytes of PRC1 or is PRC1 expression maintained in a subset, which would explain why the ovarian reserve at P5 isn’t 0? However, it wouldn’t necessarily explain why there’s a decline over time in the cKO; would that perhaps indicate a role of PRC1 in the continued maintenance of the dictyate stage?*

As shown in Fig. 1c, *Ddx4*-Cre mediated deletion of PRC1 is highly efficient (99%) in P1 oocytes. Therefore, it is unlikely that PRC1 expression is maintained in a subset of oocytes within the primordial follicle pool. One possible reason why the ovarian reserve at P5 is not zero could be attributed to heterogeneity in ovarian reserve formation. There are two regions of primordial follicle formation in the mouse ovary. Assembly of primordial follicles takes place from E17.5 to P5 in the medulla and cortex. The medullary follicles start to grow as soon as they are formed, whereas the cortical primordial follicles mature gradually over the reproductive lifespan of the animal (Zheng et al. 2014). Because *Ddx4*-cre expression initiates in fetal germ cells from E15 it is possible that a certain number of primordial follicles have formed before PRC1 is completely inactivated. Of note, we found a gradual decrease in the number of PRC1cKO primordial follicles with age, suggesting a role of PRC1 in maintaining the dictyate stage. We clarified these notions in the Results and Discussion sections (Lines 267-276).

3. Bioinformatic analyses and integration.

These are very challenging experiments and they have a high degree of reproducibility, especially the CUT&RUN. There are some clarifications that would be helpful.

i. I think a deeper analysis of the 104 MPI genes and/or other genes that might explain the potential causes that result in apoptosis (see point 1) would strengthen the study.

We further investigated regulation of 104 MPI genes with additional bioinformatics analysis and found a significant gain of H3K4me3 and a modest reduction of H3K27me3 around TSSs of the 104 MPI genes in PRC1cKO oocytes (New Supplementary Fig. 6d). These results further support a direct function of PRC1 in regulating expression of these targets. We further described functions of representative genes in 104 MPI genes (Lines 170-180). However, we were not able to determine whether these genes are responsible for apoptosis.

ii. Figure 1a – the heatmap are shown and the clusters are defined. However, it wasn't quite clear what a robust z-score is and how this relates to a false discovery rate. It would be helpful if this aspect of the higher level analysis wasn't well described in the Methods.

The Robust Z-Score Method is also known as the Median Absolute Deviation method and is similar to the Z-score method with some changes in parameters. Because means and standard deviation values are heavily influenced by outliers, the Robust Z-Score method uses medians and absolute deviation values from medians instead.

We determined False Discovery Rate (FDR) values based on DESeq2, which is a package commonly used for differential gene expression analyses before we draw the heatmap using the Robust Z-score method. Thereby, the Robust Z-score output is independent of FDR values.

We newly added descriptions of the Robust Z-score method and how we determined FDR values in the Methods section (Lines 398-402, 406-408).

iii. The central hypothesis is that altered H2K119ub is correlated with elevated H3K4me3 and increased expression of meiotic cell cycle (MPI) genes in the PRC1 cKO leading to oocyte atresia. I might have missed this, but what is the overall correlation between loss of H2AK119ub and gain of H3K4me3 across DEGs? Especially in the meiotic cell cycle genes? I think that the overlap of H2AK119ub is shown for the 104 MPIs in Figure 4f, but not what happens to the chromatin marks and expression of the 104 genes in the cKO?

As mentioned above, we found a significant gain of H3K4me3 and a modest reduction of H3K27me3 around TSSs of the 104 MPI genes in PRC1cKO oocytes (New Supplementary Fig. 6d). Loss of H2AK119ub and gain of H3K4me3 are highly correlated across DEGs, as

well as the 104 MPI genes in PRC1cKO oocytes. Expression of the 104 genes in PRC1 cKO oocytes has been shown in Fig. 3g and Supplementary Fig. 4c.

iv. Figure 3 – it would be really good with an integration into the findings from figure 1; e.g, which clusters to the differentially expressed genes belong to? A heatmap of the same clusters as in Figure 1a, would also be good to assess what the general, overall transcriptional effect is in the cKO between P1 and P5 in comparison to the changes occurring in the wild type.

We performed new analyses and confirmed that down-regulated clusters during POT in wild-types are derepressed in PRC1cKO oocytes at P5. These results are shown with a new heat map (New Supplementary Fig. 2f) and pie charts (New Supplementary Fig. 2g).

v. Some of the comparisons of chromatin modification use profiles from mESCs. This is understandable as these analyses are challenging. However, I think the authors would need to justify why mESC profiles are appropriate for the chromatin marks. Or at least discuss uncertainties, since this is a different stage of development and many of the marks show large changes during gametogenesis and preimplantation development.

Polycomb proteins are evolutionarily conserved epigenetic regulators that define cell identities by repressing non-lineage specific genes. Specifically, Polycomb complexes (PRC1 and PRC2) suppress developmental regulators when these genes do not have functions. This feature has been extensively studied in embryonic stem cells (ESCs) and in somatic development. Therefore, the Polycomb-targeted developmental genes identified in ESCs are commonly considered as “classic Polycomb targets” and this gene list is widely used in studies of other cell types including germ cells and early embryos (Liu et al. 2020; Chen et al. 2021). Accordingly, we used this gene list as a “positive control” in our analyses. This analysis validated our CUT&RUN experiments and demonstrated that PRC1 is responsible for suppressing inappropriate developmental programs and meiotic prophase program in perinatal oocytes.

Other comments:

A short introduction to PRC1 would be good.

Thank you for this suggestion. We also realized a short introduction to PRC1 would help readers. We updated the text and added such an introduction in Lines 51-59.

Line 101: reached to the -> reached the.

Thank you for catching this error; we corrected the text (Line 117).

Figure 1 – legend. Spell out abbreviations again in legend (POT, PPT). Please state how many oocytes were included in the RNA-seq analyses.

We updated the figure legend as suggested. To clarify, the RNA-seq analyses in Fig. 1 were performed using publicly available data (Shimamoto et al. 2019) noting that in the original study the numbers of oocytes used for RNA-seq were not documented. The numbers of oocytes we used for RNA-seq are provided in the Methods section.

Figure 3a – it would be really good if the authors added a brief explanation of the FDR calculations in the legend.

We added a brief description of how we obtained the FDR values in the Methods section, RNA-seq data processing part (Lines 398-400).

Figure 4c – what is the interpretation of the H3K4me3 distribution, which has shifted in cluster 2 compared to the other clusters? Define bars (e.g. 0 to 30) below the plots.

Please note that Clusters 2-4 differed in H3K4me3 localization (Cluster 2: H3K4me3 accumulation on 3'-side of transcription start sites (TSSs); Cluster 3: on 5'-side of TSSs; and Cluster 4: on the center of TSSs). We have included this description in the figure legend.

The bars below the heatmaps represent signal intensity, the numbers represent spike-in normalized reads counts. We added the definition of bars in the figure and clarified this point in the figure legend. In addition, details of the CUT&RUN data processing have been included in the Methods section (Lines 410-440).

Extended data figure 2e – it's hard to read the list of meiotic cell cycle genes. It might be better to move this list to a supplemental table?

Thank you for pointing this out. To address this point, we clearly labeled the panel of Supplementary Fig. 2e and provided a detailed gene list of each GO term from Supplementary Fig. 2d in Data S3.

Reviewer #3 (Remarks to the Author):

The manuscript by Hu et al. demonstrated that PRC1-mediated histone modification, most likely H2AK119ub, is required for exit of the gene expression program underlying meiotic prophase I (MPI) in perinatal oocytes. The authors first identify a large gene expression transition, perinatal oocyte transition (POT), between postnatal days 1 and 3. As its general repressive effect on gene expression, the author hypothesize that PRC1 play a role on the POT. In PRC1-deficient oocytes, aberrant gene expression, many of which are upregulated, was observed in P5 oocytes, and a part of these gene loci show an elevated level of H3K4me3. Importantly, genes repressed by PRC1 at POT contain meiotic genes involved in MPI. Thus, PRC1 plays a critical role on POT by repressing meiotic genes.

Using a mouse model, the authors demonstrate the role of PRC on POT, which has novelty for a better understanding gene expression regulation during early oogenesis. Importantly, genes repressed by PRC1 at POT are the genes repressed in PGCs until MPI entry, illustrating a temporal regulation of meiotic genes by PRC1. The results are clearly shown, though several points to be considered further (see below). This reviewer in principle support to publish this paper in Nature Communications.

Comments:

1. Looking at gene expression dynamics between E18.5 to P6, genes repressed by PRC1 at POT should belong to the cluster 4, as other clusters do not show clear difference in gene expression upon POT. Would it be true? For example, do 548 genes shown in Figure 3d belong to the cluster 4? Showing such information would be more informative.

We performed a new analysis that confirmed this point. As the Reviewer anticipated, Cluster 4 genes occupy nearly 80% of 548 overlap genes shown in Fig. 3d (New Supplementary Fig. 4b).

In Fig. 1a, both Cluster 3 and 4 are down-regulated during POT, especially cluster 4. Consistently, Cluster 3 and 4 genes are major groups in up-regulated DEG groups in PRC1cKO oocytes at P1 and P5 (New Supplementary Fig. 2g).

2. I cannot quite understand why the author perform CUT&RUN analysis using only P1 rather than using also P3 or P5, where difference in gene expression is clear. Given repression by PRC1, such analysis would reinforce the author's claim.

We have two main reasons for only collecting P1 oocytes for CUT&RUN analyses: 1)

Chromatin state changes usually preceded changes in gene expression. As expected, we found that differences in chromatin state between ctrl and cKO in P1 oocytes predicted differences in gene expression in P5 oocytes. 2) After P1, the numbers of small non-growing oocytes in PRC1cKO decrease dramatically, which makes it very challenging to isolate sufficient oocytes at P5 to perform CUT&RUN. Furthermore, the birth rate for PRC1 mutant females using *Ddx4-Cre* is quite low, likely a consequence of ectopic *Ddx4-Cre* expression in embryos. Because of these limitations, we were unable to acquire a high-quality CUT&RUN dataset for P3/P5 oocytes.

3. Related to the comment 2, it is not entirely clear whether the repressive effect of PRC1 is mediated H2AK119ub, as Figure 4c, which show dim signals, is the only evidence supporting the claim. Would it be possible to clear this point? Or, if impossible, the author should discuss this point more intensively.

We apologize for not providing enough sufficient information about PRC1 function. Mammalian Polycomb proteins comprise two functionally-related major complexes—PRC1 and PRC2—that catalyze monoubiquitination of H2A at lysine 119 (H2AK119ub) and trimethylation of H3 at lysine 27 (H3K27me3), respectively (Schuettengruber et al. 2017). PRCs bind primarily to gene promoters to repress transcription (Farcas et al. 2012; He et al. 2013; Wu et al. 2013; Li et al. 2017). It is well accepted that Polycomb-mediated repression is associated with these specific post-translational histone modifications, and therefore accepted that the H2AK119ub profile reflects PRC1 activity *in vivo*. We have added a short introduction about PRC1 function in the Introduction section (Lines 51-59).

In Fig. 4c, although the signal intensity of H2AK119ub looks dim in the heatmap, we detected clear enrichment of H2AK119ub around TSSs. There might be two reasons for the apparent dim signals in the heatmap: 1) Biologically, the abundance of H2AK119ub in P1 oocytes is relatively low at the TSS region because of active transcription in P1 oocytes and H2AK119ub is in a poised state; 2) Typically, for chromatin profiling experiments such as CUT&RUN or ChIP-seq, a higher amount of input leads to a high signal-to-noise ratio. Because we used an ultra-low input of oocytes for CUT&RUN due to material limitations, profiling efficiency could be limited. As shown in Supplementary Fig. 6a, at the *HoxA* cluster, the classic Polycomb targeted developmental genes, prominent enrichment of both H2AK119ub and H3K27me3 were observed, which validates our CUT&RUN experiments.

4. L168: Furthermore, we found that PRC1 directly binds to genes that remain highly expressed in PRC1cKO oocytes at P5, e.g., *Mlh1* and *Taf9b*; and loss of PRC1 binding at these loci was associated with an increase of H3K4me3 in oocytes at P1 (Fig. 4a and 4b). What is the evidence demonstrating this? Figure 4a and b show histone marks but not PRC1.

We suspect that this confusion might be caused by a similar reason with the previous point. As mentioned above, it is well established that H2AK119ub is a proxy for the activity of PRC1. We revised the text and now explain about PRC1's activity (Lines 197-201).

5. Figure 1e and f are redundant. Delete f, or explain why both are shown.

We understand that Fig. 1e and f might look redundant. However, we intended to show the distribution of individual data points using scatter plots in Fig. 1e and show the trend across time points more clearly using the line chart in Fig. 1f.

References for this section:

Chen Z, Djekidel MN, Zhang Y. 2021. Distinct dynamics and functions of H2AK119ub1 and H3K27me3 in mouse preimplantation embryos. *Nat Genet* **53**: 551-563.

- Farcas AM, Blackledge NP, Sudbery I, Long HK, McGouran JF, Rose NR, Lee S, Sims D, Cerase A, Sheahan TW et al. 2012. KDM2B links the Polycomb Repressive Complex 1 (PRC1) to recognition of CpG islands. *Elife* **1**: e00205.
- He J, Shen L, Wan M, Taranova O, Wu H, Zhang Y. 2013. Kdm2b maintains murine embryonic stem cell status by recruiting PRC1 complex to CpG islands of developmental genes. *Nat Cell Biol* **15**: 373-384.
- Li H, Liefke R, Jiang J, Kurland JV, Tian W, Deng P, Zhang W, He Q, Patel DJ, Bulyk ML et al. 2017. Polycomb-like proteins link the PRC2 complex to CpG islands. *Nature* **549**: 287-291.
- Liu B, Xu Q, Wang Q, Feng S, Lai F, Wang P, Zheng F, Xiang Y, Wu J, Nie J et al. 2020. The landscape of RNA Pol II binding reveals a stepwise transition during ZGA. *Nature* **587**: 139-144.
- Schuettengruber B, Bourbon HM, Di Croce L, Cavalli G. 2017. Genome Regulation by Polycomb and Trithorax: 70 Years and Counting. *Cell* **171**: 34-57.
- Shimamoto S, Nishimura Y, Nagamatsu G, Hamada N, Kita H, Hikabe O, Hamazaki N, Hayashi K. 2019. Hypoxia induces the dormant state in oocytes through expression of Foxo3. *Proceedings of the National Academy of Sciences of the United States of America* **116**: 12321-12326.
- Wu X, Johansen JV, Helin K. 2013. Fbxl10/Kdm2b recruits polycomb repressive complex 1 to CpG islands and regulates H2A ubiquitylation. *Mol Cell* **49**: 1134-1146.
- Zheng W, Zhang H, Liu K. 2014. The two classes of primordial follicles in the mouse ovary: their development, physiological functions and implications for future research. *Mol Hum Reprod* **20**: 286-292.

REVIEWERS' COMMENTS

Reviewer #2 (Remarks to the Author):

The revisions have significantly improved the manuscript, in particular the additional analyses, text and explanations have made clarified a number of uncertainties. The answers to my original queries have all been answered, although I have a few additional, minor comments for clarity and readability for the new sections.

1. Line 57: The new introduction of PRC1 is very helpful. It should be clarified that there are two different E3 ubiquitin ligases in PRC1.

The catalytic core of PRC1 is formed by the E3 ubiquitin ligase RNF2 (also known as RING1B) or RING1 (also known as RING1A)9, which mediates H2AK119ub, the deposition of which is essential to maintain Polycomb gene repression10,11.

- Which mediates should be which mediate or either of which mediate.

2. Figure legends. A number of my requests (minor), the authors add to the Methods. Although that is helpful, it's hasn't made reading the figures any better. Adding details on number of oocytes, FDR etc in the legends as well would improve the manuscript further and avoid any confusion.

3. Line 96: H2AK119ub, a readout of PRC1 activity....please add relevant citations.

4. Line 173: the new section adds substantially to the manuscript. Would you please add citations for the various gene functions?

5. Line 187 to 190: some of the marks are explained and others aren't. I think it would add value to explain all of the chromatin marks for those not familiar with the chromatin field.

6. A follow-up question: you mention that genes involved in kidney development were expressed in the PRC1cko. In section on chromatin marks in PRC1 (line 193) the hoax cluster is mentioned in connection with PRC1 repressing inappropriate developmental programmes. Is *hoxa* involved in kidney development and can you tie together the expression of genes in other developmental programmes with the chromatin marks more formally?

7. The addition of the analysis on the 104 MPI genes and the text (line 219 to 222) are very helpful.

8. The revised section in the Discussion (lines 255 to 277) is good and opens new avenues for research.

9. The added bioinformatics methods are helpful.

10. Determine reproductive lifespan. This is used a couple of time and what is implicit is that establishment ovarian reserve determines reproductive lifespan - the rate of depletion also matters, so perhaps using a less deterministic description of the initial size of the ovarian size might be helpful?

Reviewer #3 (Remarks to the Author):

In response to comments from the reviewer, the authors appropriately revised manuscript with additional figures and text. All question are clarified in the revised manuscript. This work provides valuable information for improving our understanding of epigenetic regulation of dormant oocytes.

RESPONSE TO REVIEWERS

GENERAL COMMENTS FOR ALL REVIEWERS

We thank the reviewers for their careful consideration of our revised manuscript (NCOMMS-22-10083-A) and helpful comments, which enabled us to polish the manuscript. To address the additional points raised by Reviewer 2, we modified the text and legends as suggested. Below, are our specific comments to each Reviewer. Please note: *Reviewer comments are in italics* and **our responses are in bold**. The Reviewers' comments have not been edited.

Reviewer #2 (Remarks to the Author):

The revisions have significantly improved the manuscript, in particular the additional analyses, text and explanations have made clarified a number of uncertainties. The answers to my original queries have all been answered, although I have a few additional, minor comments for clarity and readability for the new sections.

We thank Reviewer #2 for positive comments and further guidance to improve the manuscript.

1. Line 57: The new introduction of PRC1 is very helpful. It should be clarified that there are two different E3 ubiquitin ligases in PRC1.

The catalytic core of PRC1 is formed by the E3 ubiquitin58 ligase RNF2 (also known as RING1B) or RING1 (also known as RING1A)9, which mediates59 H2AK119ub, the deposition of which is essential to maintain Polycomb gene repression10,11.

• Which mediates should be which mediate or either of which mediate.

Both RNF2, the dominant E3 ligase of PRC1, and its paralog RING1 can deposit H2AK119ub. We have modified the text to “either of which mediates”.

2. Figure legends. A number of my requests (minor), the authors add to the Methods. Although that is helpful, it's hasn't made reading the figures any better. Adding details on number of oocytes, FDR etc in the legends as well would improve the manuscript further and avoid any confusion.

We have also added these descriptions to the legends. (Figure 3 and 4 legends).

3. Line 96: H2AK119ub, a readout of PRC1 activity...please add relevant citations.

We have added relevant citations^{1,2}.

4. Line 173: the new section adds substantially to the manuscript. Would you please add citations for the various gene functions?

We have added citations for various gene functions for meiotic genes. We added new references #37-59 (Line 171-182).

5. Line 187 to 190: some of the marks are explained and others aren't. I think it would add value to explain all of the chromatin marks for those not familiar with the chromatin field.

We have added more descriptions of H3K4me3 and H3K27me3 and relevant citations^{3,4}. H2AK119ub is explained in an earlier section.

6. A follow-up question: you mention that genes involved in kidney development were expressed in the PRC1cko. In section on chromatin marks in PRC1 (line 193) the hoax cluster is mentioned in connection with PRC1 repressing inappropriate developmental programmes. Is hoxa involved

in kidney development and can you tie together the expression of genes in other developmental programmes with the chromatin marks more formally?

Yes. Hox genes encode members of the homeobox (HOX) transcription factor family that are major regulators of animal development. Hox genes are involved in various developmental programs, including kidney morphogenesis⁵ and blood vessel development⁶, which are the representative GO terms for dysregulated genes in PRC1cKO oocytes. We have added this description to the text.

7. The addition of the analysis on the 104 MPI genes and the text (line 219 to 222) are very helpful.

Thank you.

8. The revised section in the Discussion (lines 255 to 277) is good and opens new avenues for research.

Thank you.

9. The added bioinformatics methods are helpful.

Thank you.

10. Determine reproductive lifespan. This is used a couple of time and what is implicit is that establishment ovarian reserve determines reproductive lifespan - the rate of depletion also matters, so perhaps using a less deterministic description of the initial size of the ovarian size might be helpful?

We reworded the text and use “maintains” instead of “determines” (Line 45).

Reviewer #3 (Remarks to the Author):

In response to comments from the reviewer, the authors appropriately revised manuscript with additional figures and text. All question are clarified in the revised manuscript. This work provides valuable information for improving our understanding of epigenetic regulation of dormant oocytes.

We thank Reviewer #3 for positive comments and support.

References for this section:

- 1 Cohen, I., Bar, C. & Ezhkova, E. Activity of PRC1 and Histone H2AK119 Monoubiquitination: Revising Popular Misconceptions. *Bioessays* **42**, e1900192, doi:10.1002/bies.201900192 (2020).
- 2 Tamburri, S. *et al.* Histone H2AK119 Mono-Ubiquitination Is Essential for Polycomb-Mediated Transcriptional Repression. *Mol Cell* **77**, 840-856 e845, doi:10.1016/j.molcel.2019.11.021 (2020).
- 3 Santos-Rosa, H. *et al.* Active genes are tri-methylated at K4 of histone H3. *Nature* **419**, 407-411, doi:10.1038/nature01080 (2002).
- 4 Schuettengruber, B., Bourbon, H. M., Di Croce, L. & Cavalli, G. Genome Regulation by Polycomb and Trithorax: 70 Years and Counting. *Cell* **171**, 34-57, doi:10.1016/j.cell.2017.08.002 (2017).
- 5 Patterson, L. T. & Potter, S. S. Hox genes and kidney patterning. *Curr Opin Nephrol Hypertens* **12**, 19-23, doi:10.1097/00041552-200301000-00004 (2003).

- 6 Gorski, D. H. & Walsh, K. The role of homeobox genes in vascular remodeling and angiogenesis. *Circ Res* **87**, 865-872, doi:10.1161/01.res.87.10.865 (2000).